# Multivariate Pharma Technology Transfer Analysis: Civilization Diseases and COVID-19 Perspective

**DOI:** 10.3390/ijerph20031954

**Published:** 2023-01-20

**Authors:** Karol Śledzik, Renata Płoska, Mariusz Chmielewski, Adam Barembruch, Agnieszka Szmelter-Jarosz, Angelika Kędzierska-Szczepaniak, Paweł Antonowicz

**Affiliations:** 1Department of Banking and Finance, Faculty of Management, University of Gdańsk, ul. Armii Krajowej 101, 81-824 Sopot, Poland; 2Department of Business Economics, Faculty of Management, University of Gdańsk, ul. Armii Krajowej 101, 81-824 Sopot, Poland; 3Department of Logistics, Faculty of Economics, University of Gdańsk, ul. Armii Krajowej 109/111, 81-824 Sopot, Poland

**Keywords:** technology transfer, civilization diseases, pharmaceutical industry, multivariate analysis

## Abstract

The importance of studying civilization diseases manifests itself in the impact of changing lifestyles, on the number of deaths and causes of death. Technology transfer plays an important role in the prevention and treatment of these diseases. Through this, it is possible to transfer new treatments and diagnostics to clinics and hospitals more quickly and effectively, which leads to better healthcare for patients. Technology transfer can also aid in the development of new drugs and therapies that can be effective in the treatment of civilization diseases. The paper aims to evaluate the technology transfer process in the field of civilization diseases, using COVID-19 as an example of a pandemic that requires quick development and transfer of technology. To achieve the assumed goal, we propose a multivariate synthetic ratio in the field of civilization diseases (SMTT—Synthetic Measure of Technology Transfer) to analyze data from the Global Data database. We used sub-measures like SMTT_value (Synthetic Measure of Technology Transfer_value) and SMTT_quantity (Synthetic Measure of Technology Transfer_quantity) to measure technology transfer and put the data into a graph. Our analysis focuses on 14 diseases over a period of 10 years (2012–2021) and includes nine forms of technology transfer, allowing us to create a tool for analysing the process in multiple dimensions. Our results show that COVID-19 is similar in terms of technology transfer to diseases such as diabetes, cardiovascular diseases, neurodegenerative diseases, and breast cancer, even though data for COVID-19 is available for only 2 years.

## 1. Introduction

Health issues are among the most relevant from a human perspective. In this context, the ideal that humans aspire to is health, defined by the World Health Organization (WHO) [1] as “a state of complete physical, mental and social well-being and not merely the absence of disease or infirmity”. Health is not only a matter of individual concern but also an important social issue. The importance of health is evidenced, among other things, by the establishment of the World Health Organization in 1948, which, as it declares: “leads global efforts to expand universal health coverage. We direct and coordinate the world’s response to health emergencies. Moreover, we promote healthier lives—from pregnancy care through old age” [2]. The World Health Organization’s activism regards many health problems observed worldwide for a long period, but it became particularly evident during the COVID-19 pandemic, officially declared on 11 March 2020 [3]. The enshrinement of the right to medical care in the Universal Declaration of Human Rights indicates the relevance of the health topic. Article 25, paragraph one states: “Everyone has the right to a standard of living adequate for the health and well-being of himself and of his family, including food, clothing, housing and medical care and necessary social services, and the right to security in the event of unemployment, sickness, disability, widowhood, old age or other lack of livelihood in the circumstances beyond his control” [4]. Finally, health was included in the list of 17 Sustainable Development Goals announced by the United Nations in 2015. It is directly relevant to Goal 3, which reads: “Ensure healthy lives and promote well-being for all at all ages” [5]. Therefore, health issues, including combating diseases, also diseases caused by modern society, are a global concern.

A disease is defined as “any harmful deviation from the normal structural or functional state of an organism, generally associated with certain signs and symptoms and differing in nature from physical injury” [6]. Different disease groups have been written about in the literature. One of the most traditional approaches to grouping illnesses is based on their division into mental and physical illnesses [7,8]. However, this division is not entirely sharp, as it is worth noting that links between mental and physical disorders are recognized in the contemporary, holistic approach to humans [9]. There is even an emerging view that the co-occurrence of somatic and mental disorders is the rule rather than the exception [10]. Another division of diseases speaks of infectious and non-infectious diseases. Infectious diseases refer to diseases caused by bacteria, viruses, fungi, and biologically active substances produced by them. Their characteristic feature is the possibility of transmission of germs from one individual to another and the induction of the same or similar disease phenomena in the infected person. In turn, non-communicable diseases are defined as diseases that are not transmissible directly from one person to another [11]. Many of these can be described as chronic diseases, i.e., diseases that have one or more of the following characteristics: they are long-lasting or persistent; their cause, course, and treatment are not clearly defined; they leave dysfunction when they pass and may require rehabilitation, surveillance, observation, or care [12].

Finally, one of the groups distinguished in contemporary disease classifications are civilization diseases. Their concept is based on the assumption that they are globally and commonly occurring diseases whose development and spread are caused by civilizational changes (increasing industrialization, urbanization, environmental pollution, stressful work and life environment) and human behavior (poor nutrition, low physical activity) [13]. Given the latter, a related term, lifestyle diseases, is also emerging in this context [14]. This category of diseases includes, among others: obesity, diabetes, hypertension, heart disease, cancer, allergies, but also depression [14,15,16]. Civilization diseases are estimated to be responsible for more than 80% of deaths worldwide [13]. Undoubtedly, these are diseases of particular social importance, as they affect large groups and cause high social costs [17]. The situation is somewhat different for so-called rare diseases (also known as orphan diseases)—although there are many of them (around 8000), and they affect jointly around 6 to 8% of the world’s population, a single one affects a relatively small group of people [18]. Despite the ability to treat these diseases, they are often relatively difficult to diagnose and, if diagnosed, require extremely expensive medicines. The high cost of such therapies is due not only to the high cost of conducting research but also to the relatively small potential market [19]. That is why, from a technology transfer standpoint, orphan diseases are not a priority for pharmaceutical companies as they cannot bring the expected rate of return. Obviously, when a drug is introduced for this group of diseases, the therapy price is very high—as in the case of Zolgensma, where the manufacturer set the retail price of one drug dose at USD 2.125 million [20].

The WHO has addressed the multitude and diversity of diseases by publishing a new 11th version of its classification of diseases and other health problems in 2018. This classification [21], which is used for accurate statistics, consists of more than 20 categories of diseases, as well as other health issues. These include certain infectious or parasitic diseases; cancer; diseases of the blood or blood-forming organs; diseases of the immune system; endocrine, nutritional or metabolic diseases; mental, behavioral or neurodevelopmental disorders; sleep and wakefulness disorders; diseases of the nervous system; diseases of the visual system; diseases of the ear or mastoid; diseases of the circulatory system; diseases of the respiratory system; diseases of the digestive system; diseases of the skin; diseases of the musculoskeletal system or connective tissue; diseases of the genitourinary system; conditions related to sexual health; pregnancy, childbirth or puerperium; certain conditions arising during the perinatal period; developmental anomalies; symptoms, signs or clinical findings not elsewhere classified; trauma, poisoning or specific other effects of external agents. The WHO has also given attention to other causes of health problems and death, and left room for new categories of diseases to be added in the future [22].

Pharmaceutical companies play a vital role in preventing and controlling various types of diseases alongside healthcare facilities. The industry is unique in that it must strike a balance between the economic objectives set by pharmaceutical companies, which are, after all, operating for profit, and the social expectations focused on the social mission of these entities [23]. One way to assess the activities of the pharmaceutical industry from a social perspective is to analyze the scale and direction of its technology transfer process. In this research, we focus on the activity of the pharma sector in the technology transfer process in the field of civilization diseases. Due to the separation of data and availability in the database, we decided to include COVID-19 in the group of civilization diseases. The arguments for this decision are threefold. Firstly, similarly to other civilization diseases, there is a high mortality rate due to COVID-19 and long-term complications after recovery (so-called long COVID-19) [24]. Secondly, due to the development of civilization, people’s mobility has risen exponentially and as happened in 2020, highly infectious diseases can cause a global pandemic in a very short time and with severe consequences. It is assumed, according to the literature (regarding unexpected short-term events with severe long-term results) that due to mobility-related factors, pandemics will be more frequent [25,26]. Thirdly, pandemics usually cause (like in the case of COVID-19) long-term negative effects on health and raise the probability of succumbing to civilization diseases because of complications after a pandemic disease [27].

Taking these factors into consideration, the primary purpose of this paper is to evaluate the technology transfer process from a civilization disease and COVID-19 perspective. The study also aims to evaluate COVID-19 as a potential candidate for civilization diseases from a technology transfer standpoint. In order to achieve this goal, a synthetic measure of technology transfer in the field of civilization diseases has been proposed. The study period is a decade—from 2012 to 2021.

## 2. Civilization Diseases and Technology Transfer Landscape

The importance of studying civilization diseases was first recognized in the 1970s when the impact of changing lifestyle on the number of deaths and causes of death was observed. Furthermore, epidemiological evidence has changed due to changing human habits, behavior, and mobility. Since then, due to the increasing trend of deaths caused by civilization diseases, research on their prevention, causes, and effects has been intensified [28]. Moreover, as research from Princeton University shows [29], the probability of a pandemic with effects similar to COVID-19 is about 2% per year and rising (also due to changing lifestyles and growing mobility). If this trend continues, people born in 2000 will have an approximately 40% chance of experiencing a pandemic during their lifetime. The probability of experiencing pandemics will double every decade [29]. Therefore, it is worth testing solutions for the quick development of medicines for pandemic diseases and related technology transfer. Additionally, human behavior is changing in a way that is conducive to the rapid spread of diseases.

Now, civilization diseases are the topic of interest in many research studies [16,30,31,32,33,34,35,36,37,38,39,40,41]. Taking into account the diseases that contributed to the highest mortality among people in the world at the beginning of the 21st century, the following diseases were identified: hypertension (1.1% of all deaths), diabetes (2.7%), diarrhea (2.7%), HIV and AIDS (2.7%), cancer of the bronchi and lungs (2.9%), tuberculosis (3.1%), lower respiratory tract disease (5.5%), chronic obstructive pulmonary disease (5.9%), stroke (12%), and coronary artery disease (13.2% of all deaths). The World Health Organization estimated that 18.3 million people worldwide died due to cardiovascular diseases in 2019. Considering all types of cancer, they are the second most common cause of death (after cardiovascular-coronary diseases)—the number of deaths from all types of cancer worldwide in 2019 was 10 million, with 23 million cancer cases. The five main types of cancer causing the most significant harm to patients were: tracheal, bronchial and lung cancers; colon and rectal cancer; stomach cancer; breast cancer, and liver cancer [42]. Before the outbreak of the COVID-19 pandemic, all of these diseases accounted for more than 50% of all disease-related deaths (Statista, 2022). Stroke is the third leading cause of death worldwide, after heart disease and cancer. It is also the most common cause of permanent disability in people over the age of 40—about 5 million people died from it in 2019.

However, various diseases and types of human behavior are also indicated as causes of death. The biggest killers in the world in 2019 were: passive smoking (responsible for 1.30 million deaths in the world in 2019), low birth weight (1.84 million deaths), a diet low in whole grains (1.89 million deaths), a diet high in sodium (2.31 million deaths), alcohol consumption (2.44 million deaths), obesity (5.02 million deaths), high blood sugar (6.50 million deaths), air pollution (6.67 million deaths), smoking (7.69 million deaths), high blood pressure (10.85 million deaths). In total, the listed factors were responsible for the deaths of over 46 million people worldwide in 2019 [43].

At the end of 2019, a new factor appeared that had a significant impact on both morbidity and mortality—it was the COVID-19 pandemic. The outbreak of the pandemic caused both a sharp increase in the number of cases and, consequently, in the number of deaths caused by the virus. By the end of 2020, approximately 84 million people had COVID-19 worldwide, of which approximately 2 million patients died. By the end of 2021, the number of cases had increased to 291.5 million people, with 5.5 million deaths. By October 2022, the total number of cases was over 625 million, and the number of deaths had exceeded 6.5 million. Such a sharp increase in morbidity and deaths caused by the COVID-19 pandemic has changed the perception of risk factors that may contribute to human death. In 2021, according to the opinions of respondents around the world, the ten most prominent health problems faced by people in the world were indicated as coronavirus (70% of respondents), cancer (34%), mental health (31%), stress (22%), obesity (19%), diabetes (13%), drug addiction (13%), alcohol abuse (11%), heart disease (11%), smoking (9%) [43]. It seems that the research directions undertaken by individual pharmaceutical companies are significantly influenced by people’s perception of risk and the mortality caused by particular diseases. Without a doubt, the rapid increase in the number of COVID-19 cases should result in increased interest in research on measures to counteract the disease that rapidly spread in a given period. It should be noted that the number of deaths caused by the 3-year-long COVID-19 pandemic (confirmed 6.5 million deaths as of October 2022, not including subsequent deaths from its side effects) is not as high as the annual number of deaths caused by the most common causes of human death in the world: cardiovascular diseases (18.3 million deaths worldwide in 2019), all types of cancer (10 million deaths in 2019), and slightly exceeds the annual number of deaths from stroke (5 million people in 2019). Undoubtedly, the frequency of occurrence and mortality caused by a given disease is one factor that determines the pharmaceutical industry’s interest in searching for drugs for a given disease and technology transfer.

### 2.1. Technology Transfer Landscape

Technology transfer plays an important role in the prevention and treatment of civilization diseases. Thanks to it, new treatments and diagnostics can be transferred to clinics and hospitals more quickly and effectively, leading to better healthcare for patients. Technology transfer can also help in the development of new drugs and therapies that can be effective in treating civilization diseases and also plays an important role in preventing civilization diseases by enabling wide access to information on a healthy lifestyle and disease prevention. All this makes it an important tool in the fight against civilization diseases.

Research on technology transfer began in the early 1970s and has evolved to include such areas as: the macro level of technology transfer with triple and quadruple helix [44,45,46,47], entrepreneurial ecosystems [48,49], and national systems of innovation [50,51]. Research on technology transfer also encompasses various areas such as diseases, medicine, and pharmacy [52,53,54,55,56,57,58,59,60,61]. There are relatively fewer publications on the methodology for evaluating technology transfer [62,63,64,65]. Efficient technology transfer in the value chain can be crucial in the diffusion of new innovations and technology-associated knowledge [66]. In an innovation-driven economy, technology transfer is a topic that is subject to numerous debates. It is a complex process with many aspects and blurred boundaries between knowledge transfer, commercialization, and innovation implementation [67]. Many methodological approaches have been applied in the development of knowledge about technology transfer. Yet, so far, there is no uniform, universally recognized definition of the concept or a recognized tool for measuring this process. This state of knowledge allows researchers to propose their own solutions for evaluating processes related to technology transfer [68,69,70,71].

Regarding geography, the most significant values of technology transfer in 2012–2021 occurred in North America and Europe. The results are highlighted in Table 1. In North America, the value of technology transfer transactions primarily relates to the treatment of diseases such as lung cancer (16.7% of the total value of technology transfer transactions in this region), breast cancer (16.6%), and obesity (16.4%).

In Europe, however, the most significant values of technology transfer transactions were observed for cardiovascular diseases (26.5% of the sum of technology transfer transaction values in this region), diabetes (20%), and breast cancer (19.5%). The distribution of the transaction value in Asia is more similar to that in Europe, and is as follows: cardiovascular diseases (23.4%), diabetes (20.7%), and lung cancer (10.2%). In the Middle East, the most significant values of technology transfer transactions were observed for cardiovascular diseases (21% of the sum of technology transfer transaction values in this region), obesity (18.1%), and diabetes (16.9%). South and Central America is an area where technology transfer transactions were observed for the treatment of diseases such as: diabetes (32.4% of the sum of technology transfer transaction values in this region), cardiovascular diseases (17.9%), and obesity (11.4%). The lowest technology transfer transaction values in the period 2012–2021 were observed for bladder cancer in Asia, Europe, and the Middle East (0.6%, 0.004%, and 0.05% of the sum of technology transfer transaction values, respectively), depression in North America (0.9%) and neurodegenerative diseases in South and Central America (0.07%).

## 3. Methodology

In this study, the following research methodology framework was used (see Table 2.) The research questions (RQ) were raised within the context of literature reviews on COVID-19, civilization diseases, and technology transfer:RQ1: What is the place of COVID-19 in relation to other civilization diseases in the technology transfer process?RQ2: Which civilization diseases have the highest importance in terms of technology transfer?

The development pattern method used in this research is one of the methods allowing the ranking of observations in terms of the level of phenomenon complexity, from “the best” to “the worst” according to the adopted general criteria. Depending on the complexity of the problem, the pattern of proceeding in this analysis may vary [72,73].

The research sample is unbalanced, which is a common problem in pharmaceutical and medical research [74]. It is difficult to collect full-scope data, therefore for some regions in our study, we observed oversampling, and for others, undersampling. That is why full generalization is not possible in our research. To address the problem of data imbalance, there is some related work on data augmentation and data compensation approaches [75] that focus on this topic. However, even though the research results are based on an imbalanced sample, they cover a vast number of possible observations and provide impactful insights.

The analysis was conducted on data from the years 2012–2021. The geographical scope covered such regions as: Asia, Europe, the Middle East, North America, and South and Central America. There was no available data for Africa and Australia. The source of the data was the Global Data database—module “Pharma”. The study was conducted from May to November 2022. A total of 22 sub-variables characterizing technology transfer in the field of the analyzed disease units were collected for analysis. Based on the collected data, a general market characterization was carried out mainly based on a value criterion (volume of transactions), geographical area, and the form of technology transfer. Due to the complex technology transfer process characterized by many sub-variables, it was considered reasonable to use multivariate comparative analysis to synthesize the studied phenomenon. In the next step, variables describing the technology transfer process (grant, partnership, asset transaction, contract service, acquisition, licensing, private equity, venture financing, debt offering, merger, equity offering) were divided into two groups, which characterize the process in terms of value (one group) of transaction and quantity (second group). After analyzing the gaps in the matrices, eight indicators were used for further research (Contract service and Mergers were eliminated). According to the Global Data—module “Pharma” database, the following civilization diseases were selected for research: cardiovascular diseases, diabetes, COVID-19, breast cancer, neurodegenerative diseases, lung cancer, obesity, AIDS, allergies, prostate cancer, depression, pancreatic cancer, bladder cancer, and chronic obstructive pulmonary disease.

Based on the views presented in the literature, the verification of the level of correlation between variables was abandoned [76,77]. Instead, all variables were considered stimuli, i.e., whose higher value is desirable from the point of view of the analyzed phenomenon (example: the higher the expenditures on grants, the better). As a result, three matrices were constructed. The first included all the sub-variables (both in value and quantity terms). The purpose of creating this matrix was to determine a synthetic variable (SMTT) describing the studied phenomenon and to allow ranking of the analyzed disease units according to the adopted criterion.

The second matrix was based on value indicators, while the third matrix was based on quantitative indicators. The second and third matrices were constructed to create two rankings. The first allowed ranking the disease units (from the best to the worst) in terms of the value of technology transfer transactions (SMTT_value). The second ranked the studied phenomenon in terms of the number of transactions concluded (SMTT_quantity).

These rankings allowed for a graphical presentation of the research in a coordinate system, where the *X*-axis was used to place the values of the synthetic value measure (SMTT_value), while the *Y*-axis was used to place the values of the synthetic quantitative measure (SMTT_quantity). The intersection of the coordinate system is set at the SMTT quantity mean and the SMTT_value mean. In this way, a map showing the position of each disease unit was obtained.

The study was based on the following procedure [78]:Creating a matrix of objects and features
(1)$$X=[X_{ij}]\;\left(i=1,\ldots ,n;\;j=1,\ldots ,m\right)$$
where:

*X_ij_*—the value of the *i*-th object (disease entity) of the *j*-th feature (indicator)
2.The second step of the calculation was to bring the different variables to comparable levels with standardization. As a result of diagnostic standardization, each variable will have a mean value of 0 and a standard deviation equal to 1. Standardization was made according to the following formula:(2)$$z_{ij}=\frac{x_{ij}-{\bar{x}}_{j}}{s_{j}}\;\left(i=1,\;\ldots ,\;n;j=1,\;\ldots ,\;m\right)$$where:

$z_{ij}$—standardized value of the *j*-th feature of the *i*-th object

${\bar{x}}_{j}$—the arithmetic mean of the *j*-th feature

$x_{ij}$—the value of the *j*-th feature of the *i*-th object

$s_{j}$—standard deviation of the *j*-th feature
3.The third step was to estimate the *z*_0_—Positive Development Pattern (PDP)–by setting the maximum value for stimuli and the minimum value for destimuli in each column of standardized features. Again, all variables adopted were assumed to be stimuli.(3)$$z_{0}=\left[z_{01},\;z_{02},\;\ldots ,\;z_{0m}\right]$$where:(4)$$z_{0j}=\{\begin{array}{c} \;\underset{i}{\max}\;z_{ij},\;\mathrm{when}\;\mathrm{the}\;\mathrm{variable}\;z_{j}\;\mathrm{is}\;a\;\mathrm{stimulant}\;\\ \;\underset{i}{\min}\;z_{ij},\;\mathrm{when}\;\mathrm{the}\;\mathrm{variable}\;z_{j}\;\mathrm{is}\;a\;\mathrm{destimulant} \end{array}$$

At the same time, the so-called negative pattern (anti-pattern) with the “worst” values of the variables is determined as the $z_{-0j}$ Negative Development Pattern (NDP)
(5)$$z_{-0}=\left[z_{-01},\;z_{-02},\;\ldots ,\;z_{-0m}\right]$$
(6)$$z_{-0j}=\{\begin{array}{c} \underset{i}{\min}\;z_{ij},\;\mathrm{when}\;\mathrm{the}\;\mathrm{variable}\;z_{j}\;\mathrm{is}\;a\;\mathrm{stimulant}\;\\ \;\underset{i}{\max}\;z_{ij}\;\mathrm{when}\;\mathrm{the}\;\mathrm{variable}\;z_{j}\;\mathrm{is}\;a\;\mathrm{destimulant}\; \end{array}$$
4.The next step was calculating the distance of each object from the PDP, taking into account the impact of various strength characteristics on the studied phenomenon. The formula used to determine this distance is the Euclidean distance:
(7)$$d_{i0}=\sqrt{\sum_{j=1}^{m}{\left(z_{ij}-z_{0j}\right)}^{2}}\;i=1,\;\ldots ,\;n$$where: $d_{i0}$—Euclidean distance of the *i*-th object (disease entity) from the Positive Development Pattern (PDP). The more similar to the pattern (less distant from it) an observation is, the higher the level of phenomenon complexity for that observation.

The distances so determined refer to the maximum possible distance, which is the distance between the pattern and the anti-pattern,
(8)$$d_{0}=\sqrt{\sum_{j=1}^{m}{\left(z_{0j}-z_{-0j}\right)}^{2}}$$
where: $d_{0}$—distance between the development pattern and the anti-pattern

Because the synthetic variable defined by Equation (4) is not normalized, $d_{i0}$ the ratio must be changed in the normalization process. This will lead to changes in variable preferences, where a larger value will correspond to a higher level of the studied phenomenon (competitive position). The synthetic variable will take values in the range from 0 to 1. The formula used was as follows:(9)$$SMTT_{i}=1-\frac{d_{i0}}{d_{0}}\;i=1,\;\ldots ,\;n$$
where:

SMTT*i*—Synthetic Measure of Transfer Technology—the development measure for the *i*-th object

The development measure is constructed so that it satisfies the following properties:The measure values are contained in the interval [0, 1], with the development measure calculated for a development pattern equal to one and for an anti-pattern equal to zero.The higher the level of the phenomenon, the higher the value of the measure.

In the next step, three SMTT (Synthetic Measure of Technology Transfer) indicators were determined using the above procedure. The first indicator (SMTT) was determined for all acquired variables (transaction value and number of transactions). The second indicator (SMTT_value) was determined for the transaction value of different technology transfer forms. The third indicator (SMTT_quantity) was determined for the number of transactions carried out for each form of technology transfer. The determination of SMTT_value and SMTT_quantity made it possible to compile them in the coordinate system. This allowed for the interpretation of the multidimensional position of a given disease concerning the value of the type of technology transfer transactions and the number of transactions.

The interpretation of the graphical presentation in a coordinate system (four quadrants), where the *X*-axis was used to place the values of the synthetic value measure (SMTT_value), while the *Y*-axis was used to place the values of the synthetic quantitative measure (SMTT_quantity) is as follows:Quadrant 1—relatively high values of technology transfer transactions (SMTT_value) and relatively high number of transactions (SMTT_quantity).Quadrant 2—relatively low values of technology transfer transactions (SMTT_value) and relatively high number of transactions (SMTT_quantity).Quadrant 3—relatively low values of technology transfer transactions (SMTT_value) and relatively low number of transactions (SMTT_quantity).Quadrant 4—relatively high values of technology transfer transactions (SMTT_value) and relatively low number of transactions (SMTT_quantity).

The origin of the coordinate system was determined as the average SMTT_value (0.2490), and the average SMTT_quantity (0.2890) (see Figure 1). This made it possible to obtain four quadrants of the coordinate system and group diseases into those with high and low SMTT_quantity and high and low SMTT_value relative to the average.

## 4. Results

In a one-dimensional approach, lung cancer was the disease with the highest total value in terms of different technology transfer forms during the 2012–2021 period (USD 434 billion), accounting for 17% of total transaction value (see Table A1). Breast cancer came in second (USD 429 billion), while cardiovascular diseases ranked third (USD 371 billion). Diabetes had the highest number of transactions (57,898 transactions in 2012–2021; see Table A2) followed by cardiovascular diseases and AIDS respectively. When focusing on individual forms of technology transfer, for acquisition, the highest transaction value was related to breast cancer (USD 384 billion; see Table A1) This value represents about 89% of all forms of technology transfer for this disease, also being the highest value across all diseases in the 2012–2021 period. Acquisition related to lung cancer came in second (USD 381 billion), followed by acquisition related to cardiovascular diseases (USD 298 billion).

The least amount spent in the technology transfer process during this decade was related to chronic obstructive pulmonary disease (USD 11.5 billion; see Table A1). The second-to-last place was recorded for depression (USD 20.6 billion), followed by COVID-19 (USD 31 billion). In terms of different forms of technology transfer, the least amount of money was spent on private equity transactions (USD 7.4 billion). Venture financing during the study period amounted to USD 17.1 billion, while debt offerings were at USD 19 billion. The lowest number of technology transfer transactions was observed for bladder cancer (953 transactions; see Table A2), followed by chronic obstructive pulmonary disease (1883), and COVID-19 (2910). In terms of types of technology transfer transactions, the lowest number was observed for debt offerings (81), followed by private equity (104), and asset transactions (246).

Additionally, by analyzing the value and number of transactions, the one-dimensional average values of technology transfer transactions can be determined (see Table A3). From this analysis, the highest average transaction values were found for acquisitions, including 80 billion USD related to bladder cancer. The second highest average value was for asset transactions, followed by debt offerings. On the other hand, the lowest average transaction values for the 2012–2021 period were observed for grants, specifically at USD 0.33 billion in regard to pancreatic cancer.

Given the information above, it is difficult to clearly rank selected diseases in relation to selected variables. Using reference points such as the value of transactions, the number of transactions, and the forms of technology transfer transactions, it is impossible to create a single ranking of civilization diseases. The use of partial-one dimensional data and technology transfer indicators does not provide a comprehensive view of this market over the last decade. Therefore, a multivariate (multidimensional) approach is proposed in this study. This approach will evaluate the technology transfer process in the field of civilization diseases, including COVID-19, while taking into account the value, forms, and number of transactions.

The multivariate comparison analysis indicates that cardiovascular diseases, diabetes, and COVID-19 have the highest ratio of the synthetic measure of technology transfer for civilization diseases when considering the value of technology transfer transactions, form of transactions, and their number (SMTT) (see Table 3). Additionally, it is worth noting that the analyzed period is a decade and the data for COVID-19 only covers the last three years, which highlights the significance of the activities in this field.

The multidimensional position of a given disease in relation to other diseases, as viewed from the perspective of transaction values (SMTT_value) can yield varying results. However, the first and second positions are the same as in SMTT, while lung cancer ranked third. At the same time, SMTT_quantity ranked diseases in the following order: diabetes, cardiovascular diseases, and COVID-19.

By isolating the synthetic variables SMTT_value and SMTT_quantity, a tool (a coordinate system for multidimensional evaluation of the position of a variable) was created for assessing the position of a given disease in comparison to other civilization diseases.

One of the initial conclusions is that none of the civilization diseases plus COVID-19 under study were observed in the second quadrant of the coordinate system (see Figure 1). This means that during the analyzed decade of 2012–2021, there were no civilization diseases plus COVID-19 that had both low values of technology transfer transactions and a high number of technology transfer transactions simultaneously. In other words, a quadrant with a relatively high number of technology transfer transactions and relatively lower technology transfer transaction values was not observed.

**Figure 1 ijerph-20-01954-f001:**
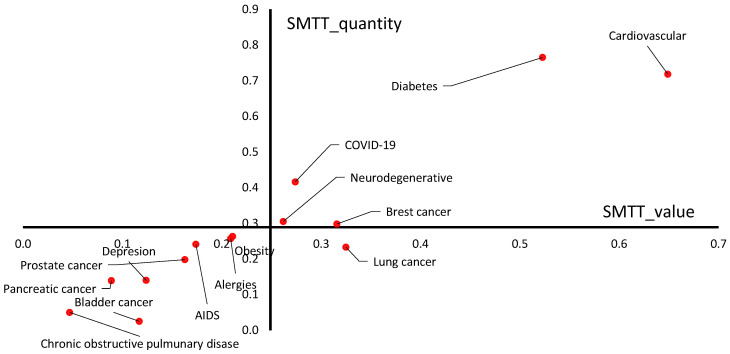
Multidimensional position of civilization diseases during the 2012–2021 period from the perspective of SMTT_value and SMTT_quantity. Source: Own elaboration based on the Global Data database—module “Pharma”.

The fourth quadrant included only one civilization disease—lung cancer (SMTT_value = 0.325; SMTT_quantity = 0.233). Therefore, this quadrant is characterized by high values of technology transfer transactions (SMTT_value) and a low number of transactions (SMTT_quantity). This means that during the research decade, lung cancer was reported to have relatively high values of technology transfer transactions and a relatively low number of these transactions simultaneously.

The least favorable multidimensional positions of civilization diseases and COVID-19 are in the third quadrant. This quadrant has relatively low values of technology transfer transactions and a relatively low number of these transactions simultaneously. As a result of the study, diseases such as: chronic obstructive pulmonary disease, bladder cancer, pancreatic cancer, prostate cancer, depression, AIDS, allergies, and obesity were observed in the third quadrant. It should be noted that obesity and allergies were located closer to the origin of the coordinate system, meaning they had a higher SMTT_value than the other diseases from the third quadrant.

The first quadrant is the most favorable from the point of view of interpreting the proposed multidimensional tool. In this section, one can observe high values of technology transfer transactions (SMTT_value) and a high number of transactions (SMTT_quantity) simultaneously. As a result of the research, four civilization diseases and COVID-19 were observed in this quadrant. The observed diseases include breast cancer, neurodegenerative diseases, diabetes, and cardiovascular diseases. Relatively, cardiovascular diseases have the highest technology transfer values and the highest number of transactions. Diabetes is in second place. COVID-19, breast cancer, and neurodegenerative diseases were observed closer to the center of the coordinate system.

## 5. Discussion

When preparing and conducting the study, as well as formulating conclusions, the authors made every effort to ensure that they were as reliable and credible as possible. However, they were aware of certain limitations that always occur with this type of activity.

First of all, in the study, we are dealing with a complex phenomenon, i.e., one that is described by more than one variable. This created the premises for synthesizing inference despite certain limitations. The main one is the unbalanced research sample we used in the data analysis process. As mentioned earlier, this makes it impossible to extrapolate the results for all the world regions. However, even though it shows the use of a multidimensional comparative analysis ratio as part of the analysis of the scope of technology transfer, it allowed for the ordering (hierarchization) of the analyzed civilization diseases and COVID-19 due to a synthetic SMTT measure aggregating information contained in 18 indicators. Because of this deficiency, we would like to encourage further scientific discussion on the topic described in this article and to broaden the analysis of technology transfers in future research.

Another important limitation is the deliberate selection of civilization diseases under analysis. As mentioned in the study, there is no clearly defined and closed catalog of civilization diseases. Therefore, the authors had to decide which disease entities would be included in the research. Moreover, by choosing the Global Data database—module “Pharma” as the primary data source, it was necessary to adopt the classification approach applied there.

A debatable element, although justified from the authors’ point of view, is the inclusion of COVID-19 in the analysis, which, as mentioned earlier, is not considered a typical disease of civilization, often associated with lifestyle. However, research confirms the relationship between susceptibility to severe COVID-19 and lifestyle diseases. In addition, the aggravation of lifestyle diseases (in particular heart disease, diabetes, depression) during the COVID-19 pandemic is confirmed. Considering the above and the fact that large financial outlays have been included in the prevention and treatment of COVID -19, placing this disease on the coordinate system is justified. In addition, given the situation related to the recent spread of the SARS-CoV-2 virus, it was decided to include in the analysis the disease entity caused by the said virus, COVID-19. The authors are aware that, due to its nature and transmission routes, it is not a civilization disease; however, they considered it advisable to show the impact of the appearance of this disease on the transfer of technologies dedicated to civilization diseases. Due to the relatively short observation period of COVID-19, resulting from the fact that this disease appeared in 2019, it is difficult to predict how long it will affect the transfer of medical technologies, including those related to civilization diseases. Moreover, the authors are also aware that the forms of technology transfer included in the study may not be complete, and other divergences may be found. Nevertheless, considering the completeness of the data in the database, it was concluded that the given number of forms of technology transfer transactions is sufficient to be tested. The last limitation that may insignificantly affect the results and conclusions of the study is the relatively few but still emerging data gaps. Therefore, it made it necessary to eliminate two forms of technology transfer in research: “mergers” and “contract service”.

## 6. Conclusions

The synthetic ratio proposed in the article allows for a certain prioritization and systematization of the level of technology transfer for various civilization diseases and COVID-19. The fact that a specific disease is located in a given quadrant of the coordinate system can be an added value from the point of view of pharmaceutical companies, the World Health Organization, doctors, and patients themselves. In addition, it also provides an understanding of the “rank” of specific diseases from the point of view of the technology transfer process. The first quadrant included the leaders—diseases characterized by a relatively high multidimensional position in terms of both the value and number of technology transfer transactions, while the third included diseases with relatively low transaction values and a relatively small number of transactions. Additionally, it can be concluded that lung cancer is closer to the first quadrant than to the third from the Euclidean distance point of view. Since COVID-19 was ranked third out of ten surveyed diseases in terms of technology transfer (according to SMTT and SMTT_quantity) within two years, it raises a fundamental question of which other diseases, including civilization diseases, may have been affected during this period.

It should be noted that COVID-19 is not considered a traditional civilization disease, but it does share some similarities. As the long-term effects of COVID-19, commonly known as long COVID-19, become more apparent, it could be considered a civilization disease, particularly in light of the increasing likelihood of pandemics every decade. The study found that COVID-19 was observed in the first quadrant of the coordinate system in a cluster with diseases such as neurodegenerative diseases and breast cancer, even though data for COVID-19 was only available for just over two years. In the first quadrant, a relatively close multidimensional location was recorded for cardiovascular diseases and diabetes, and these two diseases can be evaluated as having the highest technology transfer potential. They intersect with chronic obstructive disease and bladder cancer, which have the least activity in terms of both the value and number of technology transfer from 2012–2021. It is important to note that the research period for COVID-19 is shorter than for traditional civilization diseases, so additional research in the future may be necessary to further explore the direction proposed by the authors.

This paper provides valuable insights for further study. Firstly, a one-dimensional analysis would not have been able to yield such results. Therefore, further improvement of the multidimensional approach to assessing the technology transfer process could be the basis for future scientific research. Secondly, taking into account the changes taking place in the environment and society’s behavior, it is reasonable to conduct research on potential or new civilization diseases. Thirdly, the research problem highlights the fact that technology transfers related to both civilization diseases and COVID-19 can be used to consider the impact of one disease on the technology transfer of another. In other words, from a business perspective, which civilization diseases cease to exist and which become more attractive to the pharmaceutical industry. Additionally, it presents the results for pandemics as potential generators of additional cases and accelerators for their intensification in all the studied regions, and possibly worldwide. In the context of WHO analyses, this is an important research implication.

## Figures and Tables

**Table 1 ijerph-20-01954-t001:** Values of technology transfer transactions in the field of civilization diseases globally in the 2012–2021 period [million USD].

Disease	Asia	Europe	Middle East	North America	South and Central America
Brest cancer	6263	136,283	888	280,246	1552
Lung cancer	7765	121,208	2503	281,072	2768
Cardiovascular	17,720	185,175	5608	178,914	6081
Diabetes	15,710	139,331	4511	157,097	10,991
Obesity	5921	8186	4833	276,391	3872
Bladder cancer	495	322	137	162,844	183
Neurodegenerative	2438	5197	n.a.	122,260	27
Pancreatic cancer	1540	4314	470	95,581	55
AIDS	3674	50,472	1000	21,411	953
Prostate cancer	1619	19,338	855	33,516	975
Allergies	2823	9232	1363	36,538	1328
COVID-19	5690	12,044	666	17,250	1311
Depression	3885	5179	3770	15,805	3765
Sum	75,543	696,281	26,604	1,678,925	33,861

Source: Own elaboration based on the Global Data database—module “Pharma”.

**Table 2 ijerph-20-01954-t002:** Research framework.

Phase	Step	Tool	Outcome	Part of the Paper
Defining and designing	1. Identifying the problem	Literature review protocol	Research problem	Section 1, Section 2 and Section 2.1
	2. Describing the method	Designing data collection and analysis	Literature review protocol	Section 3
Preparing, collecting	3. Conducting a data collection and preparation	Filled data protocol	Table A1, Table A2 and Table A3	Appendix A
	4. Preparing results	Paper template	Results, Figure 1.	Section 3 and Section 4
Analyzing and concluding	5. Modifying or enhancing theory (if required)	Literature review protocol	Finished manuscript	Section 5
	6. Concluding	Research report template, paper template	Finished manuscript	Section 6

**Table 3 ijerph-20-01954-t003:** SMTT, SMTT_value, SMTT_quantity, and ranking position in the field of civilization diseases during the 2012–2021 period.

Disease	SMTT	Position	SMTT_Value	Position	SMTT_Quantity	Position
Cardiovascular	0.684	1	0.649	1	0.718	2
Diabetes	0.633	2	0.523	2	0.765	1
COVID-19	0.347	3	0.274	5	0.416	3
Brest cancer	0.306	4	0.316	4	0.298	5
Neurodegenerative	0.285	5	0.262	6	0.305	4
Lung cancer	0.274	6	0.325	3	0.233	9
Obesity	0.238	7	0.211	7	0.263	6
Allergies	0.234	8	0.209	8	0.257	7
AIDS	0.209	9	0.174	9	0.241	8
Prostate cancer	0.182	10	0.163	10	0.198	10
Depression	0.132	11	0.124	11	0.140	11
Pancreatic cancer	0.116	12	0.089	13	0.139	12
Bladder cancer	0.067	13	0.117	12	0.025	14
Chronic obstructive pulmonary disease	0.049	14	0.047	14	0.050	13

Source: Own elaboration based on the Global Data database—module “Pharma”.

## Data Availability

Not applicable.

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
