# Peer review of "Multivariate Pharma Technology Transfer Analysis: Civilization Diseases and COVID-19 Perspective"

_ijerph, 2023, doi:10.3390/ijerph20031954_

Round 1

Reviewer 1 Report

This paper is well written in general. The authors studied technology transfer process and its influences on civilization diseases, as well as the COVID-19.

Although covid-19 is a popular research topic, further explanation is necessary: why is it analyzed with the study of civilization diseases?

A Synthetic measure of technology transfer is proposed.Technology transfer potentials are evaluated for different diseases based on Global Data database, the idea is novel and the interested readers can benefit from the important idea of data-driven approaches. 

As the data distribution over different regions is unbalanced, the statistical significance of the results should be further discussed. To the data imbalance problem, some related work on data augmentation and data compensation approaches could be discussed: "Diabetes Mellitus risk prediction using age adaptation models".https://doi.org/10.1016/j.bspc.2022.104381.

Reviewer 2 Report

First of all, the quality of the English writing and style needs big improvement. Some sentences are incomplete or difficult to understand. Lots of repetitive writing. It took me quite a while to understand the authors, same for other audience. 

Specific comments:

- The abstract should be rewritten to accurately reflect the content of the paper.
- What are SMTT, SMTT_value, SMTT_quantity? Do you think the audience could understand your abbreviation in the abstract?
- What do you mean by the first quadrant in the abstract? Do you think the audance would know the meaning?
- Should you use decimal comma?
- Line 87-89 and 91-92 needs citation.
- Table 1 needs citation.
- Figure 1 needs to specifiy why the x-axis and y-axis is not at the origin.
- All equations need to be revisited. There are errors that need to be fixed and made consistant.
- You need to cut off all unnecessary sentences. For example, "The assumed research aim has been achieved. The construction of a multi-dimensional tool for assessing the technology transfer process for civilization diseases in 2012-2022 allowed for the formulation of conclusions." means nothing.
- You also need to cut off the repetitive  through out the paper.

Summary:

It is essential to improve the quality of English writing and style in order to make the manuscript more understandable and engaging for the audience. Some specific suggestions for improvement include rewriting the abstract to better reflect the content of the paper, providing more context or explanation for abbreviations such as SMTT and SMTT_value, using decimal commas consistently, properly citing relevant sources, including details about the scales used in figures, and cutting out unnecessary or repetitive sentences. These changes will help to ensure that the manuscript is clear and compelling for the intended audience.

Reviewer 3 Report

Dear Authors, 

This paper aims to evaluate the technology transfer process in civilization diseases and the COVID-19 point of view.

There is some correction required to improve the quality of this paper.

1- The problem statement is not formulated well.

2- the list of contributions of this paper is not listed.

3- why is the civilization diseases analysis important.

4- the research methodology framework is not included  

5- The evidence of data in table 1 is required.

6- The results discussion section is not enough.

Round 2

Reviewer 2 Report

The authors have addressed all comments. Minor typesetting edits required. 

Reviewer 3 Report

Well done.

I am satisfied with the corrections.

Regards